# Detection of Monkeypox Cases Based on Symptoms Using XGBoost and Shapley Additive Explanations Methods

**DOI:** 10.3390/diagnostics13142391

**Published:** 2023-07-17

**Authors:** Alireza Farzipour, Roya Elmi, Hamid Nasiri

**Affiliations:** 1Department of Computer Science, Semnan University, Semnan 35131-19111, Iran; alirezafarzipor@gmail.com; 2Farzanegan Campus, Semnan University, Semnan 35197-34851, Iran; elmi.roya79@yahoo.com; 3Department of Computer Engineering, Amirkabir University of Technology (Tehran Polytechnic), Tehran 15916-34311, Iran

**Keywords:** monkeypox, XGBoost, SHAP, MPXV, machine learning

## Abstract

The monkeypox virus poses a novel public health risk that might quickly escalate into a worldwide epidemic. Machine learning (ML) has recently shown much promise in diagnosing diseases like cancer, finding tumor cells, and finding COVID-19 patients. In this study, we have created a dataset based on the data both collected and published by Global Health and used by the World Health Organization (WHO). Being entirely textual, this dataset shows the relationship between the symptoms and the monkeypox disease. The data have been analyzed, using gradient boosting methods such as Extreme Gradient Boosting (XGBoost), CatBoost, and LightGBM along with other standard machine learning methods such as Support Vector Machine (SVM) and Random Forest. All these methods have been compared. The research aims to provide an ML model based on symptoms for the diagnosis of monkeypox. Previous studies have only examined disease diagnosis using images. The best performance has belonged to XGBoost, with an accuracy of 1.0 in reviews. To check the model’s flexibility, k-fold cross-validation is used, reaching an average accuracy of 0.9 in 5 different splits of the test set. In addition, Shapley Additive Explanations (SHAP) helps in examining and explaining the output of the XGBoost model.

## 1. Introduction

It was in the Democratic Republic of the Congo where the first human cases of monkeypox were discovered and reported as early as the 1970s [1]. In numerous nations throughout the world, unprecedented monkeypox outbreaks have been documented since May 2022 [2,3,4]. Monkeypox is a viral zoonosis that generates symptoms comparable to those experienced by people who have smallpox [5]. Infection with the monkeypox virus and the orthopox DNA virus is the main cause of the disease [6]. The orthopoxvirus genus employs diverse strategies to evade the host’s defense mechanisms, allowing the virus to enter undetected or unrecognized by the host’s systems [7]. Two distinct MPXV (Monkeypox Virus) strains are unique to Africa, with clade I predominating in central Africa and clade II in western Africa [8]. Unlike smallpox and chickenpox viruses, which can only spread from person to person through direct intimate contact with an infected person, MPXV may be spread between animals and people via blood and other body fluids [5].

Ahsan et al. [9] developed a model, using Generalization and Regularization-based Transfer Learning approaches (GRA-TLA) for binary and multiclass classification and tested the model on ten different Convolutional Neural Network (CNN) models in three separate studies. In the first and second studies, they showed that the proposed model using Extreme Inception (Xception) can separate individuals with and without monkeypox with an accuracy of 0.77 to 0.88. In the third study, they used Residual Network (ResNet)-101 with an accuracy of 0.84 to 0.99. They compared their model with TL approaches and showed that it was efficient.

MonkeyNet was created by Bala et al. [10], who used a dataset called MSID (Monkeypox Skin Images Dataset) accessible from the Mendeley Data database and developed a modified DenseNet-201 by creating a deep neural network. The accuracy of the model in the original dataset was 0.98, and in the augmented dataset, 0.98. Jaradat et al. [11] compared five pre-trained models: VGG19, VGG16, ResNet50, MobileNetV2, and EfficientNetB3. They discovered MobileNetV2 had the best result with an accuracy of 0.98. The model was tested with a different dataset and managed to achieve an accuracy of 0.94. Altun et al. [12] used the same method. Six deep-learning methods were utilized for a new problem. Best performance after optimization was reached by a method called MobileNetV3-s with an accuracy of 0.96. Kundu et al. [13] and many others have done this work using transfer learning and classifying monkeypox images. Iftikhar et al. [14] proposed a novel technique for accurate short-term forecasting of monkeypox cases. It involves filtering the time series into trend and residual subseries and utilizing machine learning (ML) models for prediction. Kumar Mandal et al. [15] proposed ML and PSO (Particle Swarm Optimization) clustering for monkeypox cases. The method is useful for forecasting and recognizing its symptoms. PSO is a bio-inspired algorithm, being a computational method to find an optimal solution [16]. Bhosale et al. [17] have done almost the same thing. They implemented a method using time-series data analysis and tried to forecast the outbreak. Khafaga et al. [18] achieved a remarkable accuracy of 0.98 in the detection of monkeypox by utilizing a deep convolutional neural network. They employed the AL-Biruni Earth radius stochastic fractal search algorithm, alongside popular deep-learning models to optimize their classification system. This performance was accomplished by leveraging an open-source dataset specifically curated for monkeypox image classification. Ahsan et al. [19] presented a model to diagnose monkeypox with the VGG16 method. They created their dataset by collecting images published on Google and using transfer learning to create a model based on VGG16 in two studies. The first study was able to distinguish monkeypox and chicken pox from each other, and the second study managed to distinguish monkeypox from other diseases (chicken pox, measles, and normal skin). They managed to get an accuracy of 0.97 in the first and an accuracy of 0.89 in the second study from their training data. Ozsahin et al. [20] utilized AlexNet, VGG16, and VGG19 in their detection process for monkeypox and chickenpox datasets, achieving a top classification accuracy of 0.99 with their proposed deep-learning model. Sitaula and Shahi [21] focused on the diagnosis of monkeypox by deep learning. They compared 13 pre-trained deep-learning models and used the best one to build their model. Their resultant accuracy in diagnosing the disease was 0.87. Saleh and Rabie [22] presented a strategy called Human Monkeypox Diagnosis (HMD), which consisted of two main parts: (1) finding the best features, using Improved Binary Chimp Optimization (IBCO), and (2) diagnosing the disease based on the found features. Finally, HMD obtained an accuracy of 0.98. Ali et al. [23] compiled and categorized a database of human monkeypox images. They employed the VGG16, ResNet50, InceptionV3, and Ensemble techniques for classification. Almufareh et al. [24] achieved 0.93 accuracy using their proposed model and various deep-learning models on open-source monkeypox skin image datasets. Sahin et al. [25] have used pre-trained deep learning using a mobile application to diagnose monkey pox. They were able to achieve an accuracy of 91.11 in image classification using deep transfer learning. According to Javelle et al. [26], a review of monkeypox outbreaks defined the disease’s clinical spectrum. The authors developed a self-administered questionnaire to track symptoms for case management, contact surveillance, and clinical studies.

This research focuses on using machine learning algorithms to detect monkeypox based on symptoms rather than images of the disease. Using images to detect diseases has limitations of its own and may not be practical in many real-world scenarios. For example, the co-occurrence of fever and rash in patients who are critically ill poses a complex diagnostic challenge [27]. Therefore, using symptoms as the basis for detection can provide a more functional and efficient solution. To achieve this, a dataset was created using published data on monkeypox. It included information on monkeypox patients’ symptoms and their diagnoses. All presented models are like a black box, and their output is not interpreted. Using SHAP, we analyzed the model’s output and solved the problem. The dataset was then analyzed using various machine learning methods, with the goal of developing a model that can accurately diagnose monkeypox based on symptoms. XGBoost, CatBoost, and LightGBM, all gradient-boosting algorithms commonly used in machine learning, were compared to each other in this study. These algorithms work by training multiple decision trees and combining their predictions to make a final prediction, a process known as boosting. Boosting can improve performance compared to using a single decision tree, making these algorithms suitable for the task. The research also compared these three algorithms with Random Forest and SVM, popular machine learning algorithms that are helpful in detecting monkeypox. The study results showed that XGBoost performed the best in terms of accuracy in diagnosing monkeypox based on symptoms. The final proposed model in this study is XGBoost, which had the highest accuracy in diagnosing diseases based on symptoms. The model has been evaluated using k-fold cross-validation for further investigation, and the model’s flexibility and reliability have been discussed. The model was also compared with other models using evaluation metrics such as precision, recall, and *F_1_-Score*. It is important to note that the use of machine learning algorithms to detect monkeypox is still in its early stages of development, and further research is needed to improve the performance of these algorithms.

Additionally, these algorithms must be accompanied by proper validation and testing to ensure they are reliable and accurate before they can be used in real-world applications. The research also suggests that, with the availability of more data, better performance can be achieved. In conclusion, this research demonstrates the potential of using machine learning algorithms to detect monkeypox based on symptoms rather than images. The proposed model, XGBoost, has shown promising results in terms of accuracy and has the potential to be used in real-world applications to improve the speed and accuracy of monkeypox diagnosis. However, further research and validation are needed to ensure the reliability and accuracy of this model before it can be used in practice. To the best of the author’s knowledge, this is the first symptoms-based model for detecting monkeypox disease. The major contributions of this study are summarized as follows:Creating a symptom-based dataset using published reports of monkeypox disease;Presenting the first model for diagnosing monkeypox based on symptoms;Using SHAP to interpret the output of the XGBoost model;Evaluation and comparison of ML models, i.e., XGBoost, LightGBM, CatBoost, Random Forest, and SVM.

## 2. Materials and Methods

### 2.1. Dataset

The dataset used in our work is published on Kaggle by ‘‘Larxel”, titled “Global Monkeypox Cases (daily updated)” [28]. It is collected by “Global Health” and used by “World Health Organization”. This dataset contains the timeline for confirmed cases with respect to date. It also contains some other details on every case that is being reported [29]. The dataset contains more than 30 fields, many of them empty. The most important of these field are Symptoms, Status, Location, City, Age, and Gender. In this dataset, symptoms are very diverse and do not have a clear structure. Furthermore, cities, and countries, have no bearing on the disease. So, from all the data, we separated only two columns, “Symptoms” and “Status”, and eventually, arrived at a new dataset based on the former. This was done by creating columns of all existing symptoms, wherein in the case of the patient having a particular symptom, we gave it a value of 1, and otherwise, a value of 0. Because the data was updated daily and without a specific structure for recording information about patients, even the same symptoms were recorded differently. For example, “Rash” is registered with other titles such as “Rashes”, “Rash on the skin”, “skin rashes”, etc., all of which are similar, and for this reason, were combined. In the end, there were about 46 columns, one ID column, 44 columns of symptoms, and a disease status column as the last one. After cleaning the data according to the mentioned steps, only 211 of them had registered and identifiable signs to make up our dataset. As mentioned, the final dataset contained only 0 and 1, and no further pre-processing was needed. The test set in all models was 20 percent.

### 2.2. Shapley Additive Explanations

The interpretability of models is a constant issue in machine learning. Lundberg and Lee’s SHapley Additive exPlanation (SHAP) is a technique for deciphering predictions made by machine learning models utilizing Shapley values [30,31]. The Shapley value for each feature represents the influence of that element on the result produced [32,33]. The formulation of the SHAP model (*g*(*z*)) is determined by the linear sum of input features, which is as follows:(1)$$f\left(x\right)=g\left(z^{\prime }\right)=\phi_{0}+\sum_{i=1}^{M}\phi_{i}{z\prime }_{i}$$
where z′ is the simplified input, $\phi_{0}$ is a constant value, M is the quantity of attributes, and ${z\prime }_{i}$ is whether the ith characteristic is noticed or not [34,35].

Given a model f and the Shapley values $\phi_{i}$, Equation (2) allows determination of the value of each input characteristic: (2)$$\phi_{i}=\sum_{S\epsilon N\backslash \{i\}}\frac{\left|S\right|!\left(M-\left|S\right|-1\right)!}{M!}[f(S\;\cup \;\{i\})-f(S)]$$
where *S* is a collection of indices in $z^{\prime }$, which is not zero, and N denotes the collection of all input characteristics [32,35,36].

### 2.3. Extreme Gradient Boosting

XGBoost is an optimized distributed gradient boosting toolkit that has been built to be effective, adaptive, and portable [36,37]. Chen and Guestrin created the XGBoost algorithm in 2016 [38]. It offers parallel tree boosting and is an improved variant of the GBDT (Gradient Boosted Decision Tree) approach (also known as GBM) [39]. The model’s anticipated output $\hat{y}$ can be calculated using an input feature vector *x* = [*x*_1_, *x*_2_, …, *x*_n_] ^T^ as follows:(3)$$\hat{y}=\sum_{k=1}^{K}f_{k}\left(x\right),\;f_{k}\;\in \;\Gamma$$
where K stands for how many weak learners there are. The weak learner’s hypothesis space, Γ, represents the function $f_{k}\left(x\right)$, which is a prediction score [40,41,42].

### 2.4. Support Vector Machine

The Support Vector Machine (SVM) is a popular statistically based supervised machine learning technique that is used for classification and regression problems [43]. It was developed in 1995 by Cortes and Vapnik to improve class separation and reduce prediction error. The ability of SVM to handle both linear and non-linear data is widely recognized and is highly good at overcoming dimensionality-related problems [44]. It is especially effective with high-dimensional feature spaces and limited datasets. SVM separates training data into discrete groups when working with linear data by locating a hyperplane with the greatest margin. The n-1-dimensional hyperplane and support vectors, which are locations that are most closely related to the margin edge, are also measured to calculate the most significant distance [45]. The mathematical equation for maximizing the margin is represented by Equation (4) (weights, input, and bias are denoted by *w*, *x*, and *b* [46]):(4)$$minimize=\frac{1}{2}||w|{|}^{2}\;subject\;to\;y_{i}(\langle w^{T}\;.\;\;x_{i}\rangle +b)>1\;for\;all\;i=1,\;2,\;\ldots ,\;n$$
where $y_{i}$ is the label of the i-th data point, $x_{i}$ is the *i*-th data point, and n is the number of data points. The constraint ensures that all data points are on the correct side of the decision boundary [47].

SVM uses the kernel trick and several kernel functions to determine the optimum hyperplane to linearly divide the data when dealing with non-linear data [48]. Below is a list of all of the possible kernel functions that were investigated in this study to determine which ones were the best [46].

Equation (5), where c is a constant integer, represents the Linear Kernel function.
(5)$$K\;\left(x_{i},x_{j}\right)=x_{i}^{T}x_{j}$$

Specifically, the Polynomial Kernel function is represented by Equation (6), where *d* denotes the polynomial’s degree, γ is its slope, and *r* represents a constant factor.
(6)$$K\;\left(x_{i},x_{j}\right)=({\gamma x}_{i}^{T}x_{j}+r{)}^{d},\;\gamma >0$$

The RBF Kernel function is represented by Equation (7), where $exp(-\gamma \;||x_{i}-x_{j}^{2}||\;)$ is the Euclidean distance between two points $x_{i}$ and $x_{j}$ and γ is the Gamma.
(7)$$K\;(x_{i},x_{j})=exp(-\gamma \;||x_{i}-x_{j}^{2}||\;),\;\gamma >0$$

The Sigmoid Kernel function is represented by Equation (8), where γ is the slope and r is a constant term.
(8)$$K\;\left(x_{i},x_{j}\right)=tanh({\gamma x}_{i}^{T}x_{j}+r)$$

### 2.5. Random Forest

Random Forest is a machine learning algorithm based on ensemble learning approaches used for classification and regression [49,50]. The Random Forest with bootstrap aggregation method is presented by Breiman [51]. Bagging helps reduce the amount of variation in the estimated predictive function. The bagging method [52] generates many subsets of the training data set, each of which is then utilized for training a classification tree on its own. The final result is determined by averaging the predictions of all the trees [53].

The expected output $\hat{\tau }\left(x\right)$ of the RF model can be calculated formally as follows for the given input feature $vectorx={\left[x^{1},\;x^{2},\;\ldots ,x^{n}\right]}^{T}$:(9)$$\hat{\tau }\left(x\right)=\frac{1}{B}\sum_{b=1}^{B}{\hat{\tau }}_{b}\;(x)$$
where B is the number of trees and ${\hat{\tau }}_{b}\left(x\right)$ indicates the estimation that the bth tree provides [54,55,56].

### 2.6. CatBoost

Engineers at Yandex proposed the gradient boosting decision tree (GBDT)-based machine learning technique known as CatBoost in 2017 [57,58]. It reduces the over-fitting of training by optimizing GBDT [59]. GBDT was created by Friedman [60].

The GBDT model uses more trees to make more accurate predictions and uses a loss function to measure how accurate the predictions are [61].

Owing to its strong performance, CatBoost has been applied in a variety of sectors, such as diabetes prediction [62], breast tumor diagnosis [63], and the identification of driving styles [64]. The category feature is swapped out for the appropriate average label value in the traditional GBDT method. To create a decision tree, nodes are separated by the average label value. This approach is known as greedy target-based statistics, and can be described as follows [57]:(10)$$\frac{\sum_{j=1}^{P}\left[X_{j.k}=X_{i.k}\right]Y_{i}}{\sum_{j=1}^{n}\left[X_{j.k}=X_{i.k}\right]}$$

The information in features, however, is typically more extensive than in the lab. When features are forcefully represented using the average label value, conditional transfer takes place.

The supplied observational dataset $D=\left\{X_{i},\;Y_{i}\right\},\;i=1,\ldots ,\;n,\;\sigma =\left(\sigma_{1},\;\ldots ,\;\sigma_{n}\right)$ is assumed to be a permutation, and the variable $x_{\sigma_{p,k}}$ is interchangeable with any other variable [57].
(11)$$\frac{\sum_{j=1}^{P=1}\left[x_{\sigma_{p,k}}=x_{\sigma_{p,k}}\right]Y_{\sigma_{j}}+aP}{\sum_{j=1}^{P=1}\left[x_{\sigma_{p,k}}=x_{\sigma_{p,k}}\right]+a}$$
where a > 0 indicates the weight of a priori, with *P* standing for a priori. The noise derived from the low-frequency category is lessened by adding an a priori.

### 2.7. LightGBM

Based on the XGBoost technology, Microsoft published an upgraded version of LightGBM in 2017 [65]. Although both LightGBM and XGBoost [38] are capable of doing parallel arithmetic, LightGBM is superior to XGBoost due to its faster training speed and lower memory occupation, both of which help lower the communication cost of parallel learning [66]. The gradient-based one-side sampling (GOSS) decision tree algorithm, exclusive feature bundling (EFB), depth-limited histogram, and leaf-wise growth approach are the significant features of LightGBM [67]. GOSS can strike a compromise between the number of samples and the precision of the LightGBM decision tree. Downsampling will focus more during training on samples with more significant gradients because they have a more considerable influence on information gain. When there are many features in a small area, LightGBM can minimize the size of the features by using EFB to join previously mutually incompatible features with a new feature [68].

To find the best model for prediction accuracy, we compared the performance of all five models introduced in the materials and method section: XGBoost, SVM, Random Forest, CatBoost, and LightGBM. This comparison was made by evaluating the performance of each model on the same dataset using the same evaluation metrics. The suggested technique’s process diagram is shown in Figure 1. The link to the repository containing the source code of this paper is provided in the “Appendix A” section.

## 3. Results and Discussion

### 3.1. Performance Comparison

The performance of each model is measured by six key metrics: accuracy, precision, recall, *F_1_*-*Score*, sensitivity, and specificity, which are defined as follows [9,69]:(12)$$Accuracy=\frac{T_{p}+T_{n}}{T_{p}+T_{n}+F_{p}+F_{n}}$$
(13)$$Precision=\frac{T_{p}}{T_{p}+F_{p}}$$
(14)$$F_{1}-Score=2\times \frac{Precision\times Recall}{Precision+Recall}$$
(15)$$Sensitivity=\frac{T_{p}}{T_{p}+F_{n}}$$
(16)$$Specificity=\frac{T_{n}}{T_{n}+F_{p}}$$
where ${(T}_{p})$ is true positive; $\left(T_{n}\right)$, true negative; ($F_{p})$, false positive; and ($F_{n})$, false negative.

These metrics are commonly used in machine learning to evaluate the performance of classification models. To compare the performance of the models, we calculated the values of these six metrics for each model, then compare the values to see which model has the highest performance. The model with the highest performance is considered the best model for prediction accuracy. In addition to these metrics, we also used other techniques to evaluate the performance of the models. For example, we used confusion matrices, which are a way to visualize the number of true positive, true negative, false positive, and false negative predictions made by the model. Table 1 shows comparison results of five ML models. 

We performed 5-fold cross-validation to confirm the XGBoost outputs. This is a widely used technique to evaluate the performance of machine learning models, particularly when the dataset is small. In 5-fold cross-validation, we randomly shuffle the data and then partition it into five equal-sized folds or subsets. Then, we treat each fold as a separate validation dataset and use the remaining four folds as training sets. This process is repeated five times, with each fold being used once as the validation dataset. The final result of the 5-fold cross-validation is produced by calculating the average performance metrics for each validation (Table 2). This gives us an estimate of the model’s performance on unseen data. Five-fold cross-validation is a powerful technique to evaluate machine learning models, as it allows us to test the model’s performance on different subsets of the data, which reduces the risk of overfitting and increases the reliability of the results. It also allows us to estimate the model’s performance on unseen data, which is essential when evaluating its generalizability.

The metrics for XGBoost achieve 1.0 in the same train and test set split, as demonstrated in Table 1, making them superior to metrics for any other method. This means that the XGBoost model has the highest performance compared to the other models. To further visualize the performance of the models, we also present the confusion matrix for each method in Figure 2. Among all five machine learning approaches, XGBoost outperformed the rest. This is due to several factors. XGBoost is an ensemble learning method that combines several decision trees, allowing it to capture more complex patterns in the data.

Additionally, it is easier to adjust the goal function, and less feature analysis is needed when using XGBoost. This means the model can be fine-tuned to optimize its performance on the specific problem. Owing to its parallel processing implementation, XGBoost also has a relatively cheap computational cost. This study used trial and error to discover the optimal algorithm’s parameters for the used model with Table 3 showing all the selected parameters.

After XGBoost, Random Forest is the second-best model and has an accurate result. Therefore, we compared these two. Table 2 shows the 5-fold cross-validation result for these two models. Both models are based on decision trees, but they differ in how they combine the decisions made by the individual trees.

When comparing the performance of XGBoost and Random Forest, we can see that XGBoost has higher performance on averages in almost all terms. In the first validation, XGBoost has more than 4 percent, and in 5-fold cross-validation it has more than 3 percent accuracy. This means that the XGBoost model has a high performance in terms of accuracy, and also it has a good balance between precision and recall (F_1_-Score). Additionally, it has a high specificity, meaning it correctly identifies negative instances most of the time. However, it has low sensitivity, and sometimes does not correctly identify positive cases. The Random Forest model has good performance with high specificity, and a higher sensitivity than XGBoost, which serves as an advantage for it. In conclusion, XGBoost is a powerful machine learning method that performs well on the dataset used in this study.

Given the high performance and ease of use, it makes sense to choose XGBoost as a predictive model to determine SHAP values for additional investigation. SHAP values are a way to explain the output of any machine learning model, and they can help understand the contribution of each feature to the model’s prediction.

Some recent monkeypox diagnostic methodologies are presented in Table 4. These methods, including those mentioned in Section 1, operate on the basis of image data. The key distinction between these methods and the proposed approach lies in the utilization of symptoms for the purpose of diagnosis.

### 3.2. SHAP Value and Pearson Correlation

Figure 3 shows the SHAP beeswarm plot. The results in Figure 3 show that fever, skin lesions, headache, muscle pain, and rash are the most effective features for predicting the output of the monkeypox prediction model. These features have the highest importance in the model’s predictions and likely contribute the most to the model’s overall performance. The evaluation of the relationship values in the data set used in this research indicates that almost all the features have had a positive and effective relationship in the monkeypox prediction model and output production. This suggests that all the features in the data set are important and contribute meaningfully to the model’s predictions. Figure 4 and Figure 5 provide further insights into the relationships between the features and the model’s output. Figure 4 shows the mean absolute value of each feature’s SHAP values, and Figure 5 shows the Pearson correlation diagram of the data set used in the research. The Pearson correlation coefficient measures the linear relationship between two variables, ranging from −1 to 1, with 1 indicating a perfect positive correlation and −1 indicating a perfect negative correlation. It can be seen in Figure 5 that there is a significant agreement between the output of the SHAP algorithm and the Pearson correlation. This implies that the results of the SHAP analysis are consistent with the linear relationships between the features and the output. One limitation of this study is the smallness of the data set used. Although the number of recorded data is large, only a small proportion of them have symptoms that can be used in the learning model. This might have affected the generalization of the model and the validity of the results.

## 4. Conclusions

This paper proposed an ML model (i.e., XGBoost) for detecting monkeypox cases based on symptoms. To evaluate the effectiveness of the proposed method, a symptom-based dataset using published reports of monkeypox disease was created, and various ML models were compared. The experimental results indicated that XGBoost outperformed other methods, reaching an accuracy of 1.0 in the general test and 0.9 in 5-fold cross-validation. Moreover, SHAP was used to interpret the output of the XGBoost model and determine the most important parameters. The results suggested that fever, skin lesions, headache, muscle pain, and rash are the most effective features for diagnosing monkeypox. The proposed method can be easily used in all medical centers. Although the proposed model obtained promising results, the small size of the dataset is a limitation of the current study. In future research, we would like to use a larger dataset and incorporate epidemiological data into the model.

## Figures and Tables

**Figure 1 diagnostics-13-02391-f001:**
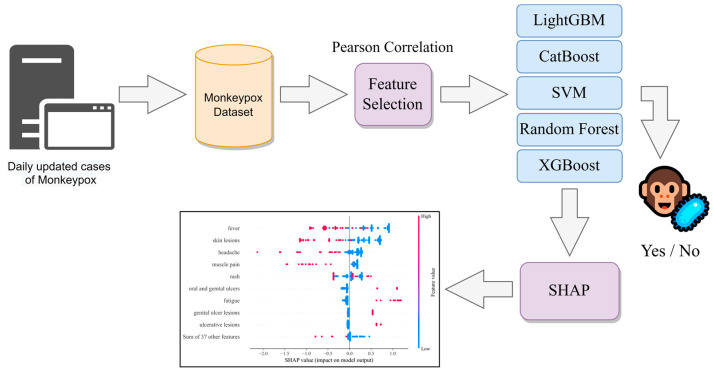
Processes diagram of the proposed method.

**Figure 2 diagnostics-13-02391-f002:**
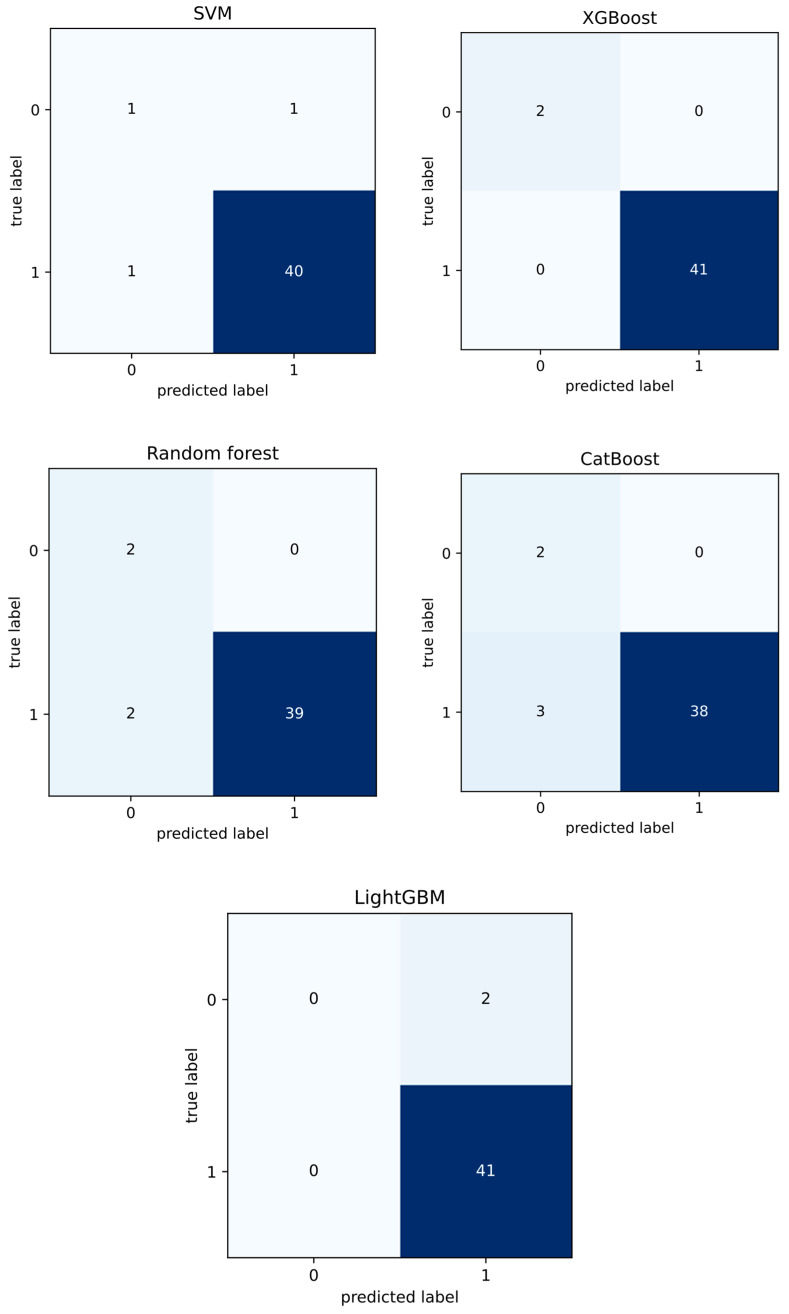
Confusion matrix for each ML method.

**Figure 3 diagnostics-13-02391-f003:**
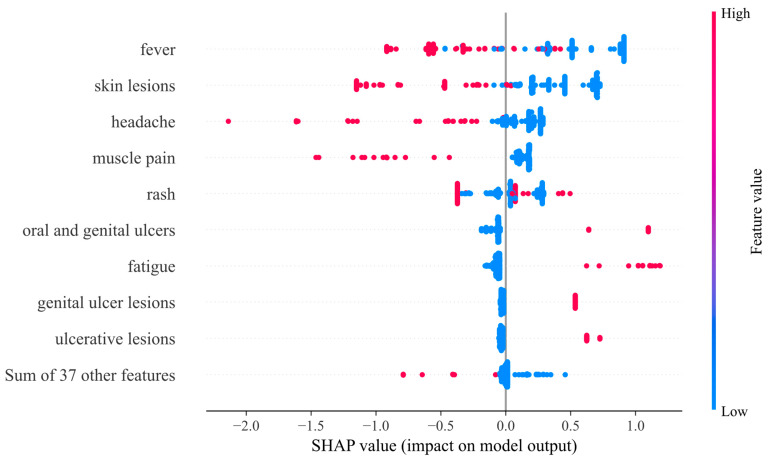
SHAP beeswarm plot (XGBoost).

**Figure 4 diagnostics-13-02391-f004:**
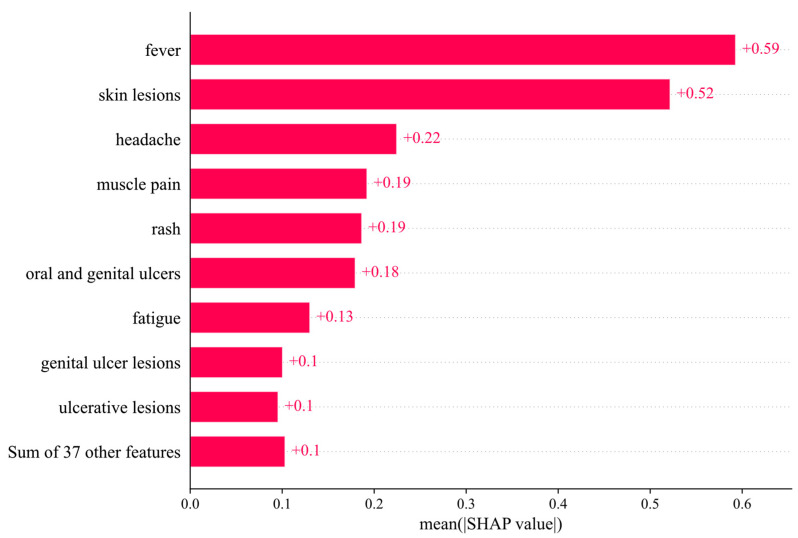
Mean absolute value of the SHAP values for each feature (XGBoost).

**Figure 5 diagnostics-13-02391-f005:**
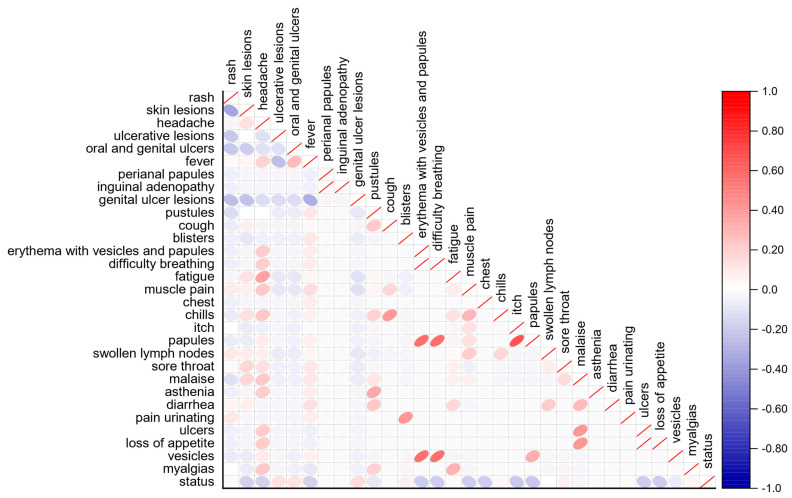
Pearson correlation.

**Table 1 diagnostics-13-02391-t001:** Comparison of five machine learning models.

Model	Accuracy	F_1_-Score	Precision	Sensitivity	Specificity
XGBoost	**1.0**	**1.0**	**1.0**	**1.0**	**1.0**
SVM	0.953	0.953	0.953	0.953	0.975
Random Forest	0.953	0.960	0.976	0.953	0.951
CatBoost	0.930	0.943	0.972	0.930	0.926
LightGBM	0.953	0.930	0.909	0.953	1.0

**Table 2 diagnostics-13-02391-t002:** Five-fold cross-validation for XGBoost.

Performance Metrics		Fold 1	Fold 2	Fold 3	Fold 4	Fold 5	Average
Accuracy	XGBoost Random Forest	0.906 0.790	0.880 0.880	0.928 0.928	0.928 0.928	0.857 0.857	**0.900** 0.877
F_1_-Score	XGBoost Random Forest	0.862 0.800	0.825 0.825	0.908 0.923	0.914 0.914	0.809 0.809	**0.864** 0.854
Precision	XGBoost Random Forest	0.822 0.811	0.776 0.776	0.933 0.921	0.933 0.933	0.878 0.878	**0.868** 0.864
Sensitivity	XGBoost Random Forest	0.906 0.790	0.880 0.880	0.928 0.928	0.928 0.928	0.857 0.857	**0.900** 0.877
Specificity	XGBoost Random Forest	1.0 0.871	1.0 1.0	1.0 0.973	1.0 1.0	1.0 1.0	**1.0** 0.969

**Table 3 diagnostics-13-02391-t003:** Model parameters.

**Parameter**	**Value**
XGBoost	
Base Learner	Gradient boosted tree
Tree construction algorithm	Exact greedy
Learning rate (η)	0.0991
Lagrange multiplier (γ)	0
Number of gradient boosted trees	80
Maximum depth of a tree	6
Minimum sum of instance weight	1
Subsample ratio of the training instances	1
Sampling method	Uniform
L2 regularization term on weights	1
Tree growing policy	Depthwise
Evaluation metrics for validation data	Negative log likelihood
SVM	
Kernel	Linear
Degree of the polynomial kernel	3
Kernel coefficient (γ)	Scale
Maximum iterations	No constraint
Shrinking heuristic	True
Probability estimates	False
Tolerance for stopping criterion	1 × 10^−3^
Random forest	
Number of trees in the forest	10
Quality of split measure function	Entropy
Minimum number of samples to split	2
Minimum number of samples at a leaf node	1
Use bootstrap samples for building trees	True
Number of jobs to run in parallel	1
CatBoost	
Number of boosting rounds	20
Learning rate	0.44
Maximum depth of a tree	5
Maximum number of trees	1000
Random seed	0
Sample weight frequency	Per tree level
Tree growing policy	Symmetric tree
Maximum number of leaves	31
LightGBM	
Number of decision trees	20
Bagging fraction	1
Number of threads in the physical core	8
Maximum depth of a tree	6
Number of boosting iterations	100
Learning rate	0.1
Maximum number of leaves on one tree	Serial
Bagging random seed	3
Dropout rate	0.1

**Table 4 diagnostics-13-02391-t004:** Monkeypox diagnostic methodologies.

Reference	Technique	Description
Haque et al. [70]	Five deep learning models such as VGG19, Xception, DenseNet121, etc.	Image-based dataset with an accuracy of 83% using Xception-CBAM (Convolutional Block Attention Module)
Sahin et al. [25]	Transfer learning methods such as MobileNetv2, GoogleNet, etc.	Image-based dataset with an accuracy of 91% using MobileNetv2
Irmak et al. [71]	VGGNet, and MobileNetV2	Image-based dataset with an accuracy of 91% using MobileNetV2
Alcalá-Rmz et al. [72]	MiniGoggleNet	Image-based dataset with an accuracy of 97%
Jaradat et al. [11]	Five pre-trained models: VGG16, ResNet50, MobileNetV2, etc.	Image-based dataset with an accuracy of 98% using MobileNetV2
Proposed Method	XGBoost	Symptom-based dataset with an accuracy of 100% using XGBoost

## Data Availability

The dataset used in this study is available at: https://github.com/alirezafarzipour/MonkeyPoxDetection (accessed on 10 June 2023).

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
