# Peer review of "Detection of Monkeypox Cases Based on Symptoms Using XGBoost and Shapley Additive Explanations Methods"

_diagnostics, 2023, doi:10.3390/diagnostics13142391_

Round 1

Reviewer 1 Report

The researchers aimed to diagnose monkeypox in this study. Gradient boosting methods such as Extreme Gradient Boosting (XGBoost), CatBoost, and LightGBM, as well as Support Vector Machine (SVM) and Random Forest methods, were used.

The title "Detection of Monkeypox Cases Based on Symptoms Using XGBoost and Shapley Additive Explanations Methods" is appropriate for the journal; however, the paper requires some revisions before it can be considered for publication.

 The contributions of this study should be provided in the introduction section.

The access dates should be added to references 24 and 25, which refer to the used data. While the dataset can be updated, the researchers should specify the date they obtained the data used in the study.

You can include the following article in the citation for performance metrics: https://doi.org/10.1016/j.compbiolchem.2022.10768

Additionally, sharing the dataset and source code on GitHub is important for the transparency of the results.

Reviewer 2 Report

To find an ML model based on symptoms to diagnose monkeypox, this study made a dataset based on the symptoms of Monkeypox. The five methods, including XGBoost, CatBoost, LightGBM, SVM and Random Forest were used to analyze the data. By comparing the accuracy, specificity, sensitivity and precision of these five methods, XGBoost was considered to be the best method in this study. I think the ML method with high sensitivity can be used for the prevention and clinical diagnosis of monkeypox infection. Laboratory testing should be the gold standard for final diagnosis. Therefore, I think there is no difference between the three methods of XGBoost, Random Forest and CatBoost (Table 1). Random Forest has better sensitivity than XGBoost(Table2).

In addition, given that some of the symptoms of monkeypox are non-specific and that epidemiology can compensate for the deficiency, epidemiological data should be included in the data.

"myalgia" is repeated in Figure 3

Reviewer 3 Report

The manuscript by Farzipour et al. entitled "Detection of Monkeypox Cases Based on Symptoms Using 2 XGBoost and Shapley Additive Explanations Methods" is a good and well-written manuscript that describes the machine learning model based on symptoms to diagnose monkeypox." The manuscript is very well written, and the analysis is excellent. I recommend it for publication after some minor corrections are made. Those minor corrections should be made to improve the quality of the manuscript. My only two comments are:

1)     Please add a diagram or figure of the processes, including XGBoost, SVM, Random Forest, Catboost, and LightGBM, to give the reader a better understanding.

2)     Please read the numerous studies (shown below), including the symptoms and action mechanisms, then compare them with your findings based on machine learning and cite them.

a)     https://www.sciencedirect.com/science/article/pii/S1477893923000194

b) https://www.nature.com/articles/s41392%E2%80%93022-01215%E2%80%934

c)     https://pubmed.ncbi.nlm.nih.gov/36298710/

d)    https://www.taylorfrancis.com/chapters/edit/10.1201/9781315099538-9/diagnostic-approach-rash-fever-critical-care-unit-lee-engel-charles-sanders-fred-lopez

Reviewer 4 Report

Farzipour et al describes application of XGBoost and Shapley Additive Explanations Methods for the Detection of Monkeypox Cases. The theme of the study is important and interesting. The authors provide qute detailed explanation of performed studies. However, some issues must be eliminated prior to publication.

1. Beginning of 3.1 section belongs to Methods.

2. The authors must provide data about models training and validation. eg trees description, perfomnce, etc. Probably such information may be presented as supplementary file.

3. Discussion section is missing any comparisons to the studies in the field. Please, provide such analysis and describe if the proposed method may be preferable to already known approaches.

4. In Conclusions the authors state that the achieved accuracy is 1, however in the text the described accuracy is 0.9.

5. Conclusions needs huge fixes. At the moment text is hard to read. Numeral unrelated sentences containes unconnected ideas. 

6. In section 2.1 please clearly indicate that 211 cases wera analyzed in this study instead of huge description of unnescessary other cases.

The paper may be published after corrction of mentioned issues.

Round 2

Reviewer 4 Report

The authors addressed all issues, the paper may be published.